# Identification of the Adversary from a Single Adversarial Example

## Abstract

Deep neural networks have been shown vulnerable to adversarial examples. Even though many defence methods have been proposed to enhance the robustness, it is still a long way toward providing an attack-free method to build a trustworthy machine learning system. In this paper, instead of enhancing the robustness, we take the investigator's perspective and propose a new framework to trace the first compromised model in a forensic investigation manner. Specifically, we focus on the following setting: the machine learning service provider provides models for a set of customers. However, one of the customers conducted adversarial attacks to fool the system. Therefore, the investigator's objective is to identify the first compromised model by collecting and analyzing evidence from only available adversarial examples. To make the tracing viable, we design a random mask watermarking mechanism to differentiate adversarial examples from different models. First, we propose a tracing approach in the data-limited case where the original example is also available. Then, we design a data-free approach to identify the adversary without accessing the original example. Finally, the effectiveness of our proposed framework is evaluated by extensive experiments with different model architectures, adversarial attacks, and datasets.

## 1 Introduction

It has been shown recently that machine learning algorithms, especially deep neural networks, are vulnerable to adversarial attacks (Szegedy et al., 2014; Goodfellow et al., 2015). That is, given a victim neural network model and a correctly classified example, an adversarial attack aims to compute a small perturbation such that the original example will be misclassified with this perturbation added. To enhance the robustness against attacks, many defence strategies have been proposed (Madry et al., 2018; Zhang et al., 2019; Cheng et al., 2020a). However, they suffer from poor scalability and generalization on other attacks and trade-offs with test accuracy on clean data, making the robust models hard to deploy in real life. Therefore, in this paper, we turn our focus on the aftermath of adversarial attacks, where we take the forensic investigation to identify the first compromised model for generating the adversarial attack. In this paper, we show that given only a **single** adversarial example, we could trace the source model that adversaries based for conducting the attack. As shown in Figure 1, we consider the following setting: a Machine Learning as a Service (MLaaS) provider will provide models for a set of customers. For the consideration of time-sensitive applications such as auto-pilot systems, the models would be distributed to ev-

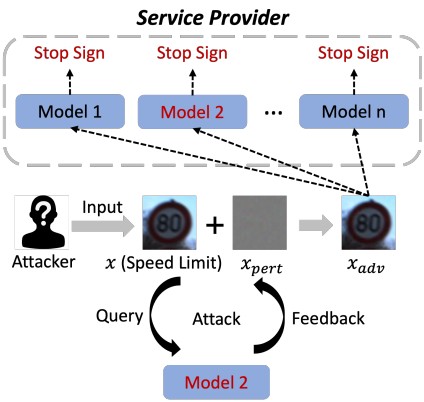

Figure 1: Illustration of the threat model in owner-customer distribution setting where the attacker conducts adversarial attacks on the assigned model 2 and uses the generated adversarial examples to attack other users' models.

ery customer locally. The model architecture and weight details are encrypted and hidden from the customers for the consideration of intellectual property (IP) protection and maintenance. In other

words, every customer could only access the input and output of the provided model but not the internal configurations. On the other side, the service provider has full access to every detail of their models, including the training procedure, model architecture, and hyperparameters. However, there exists a malicious user who aims to fool the system by conducting adversarial attacks and gaining profit from the generated adversarial examples. Since the models are trained for the same objective using the same dataset, adversarial examples generated by the adversary could be transferred to the other users' models with a very high probability, 100% if the models are the same. Thus it is critical for the interested party to conduct the investigation and trace the malicious user by identifying the compromised model. Taking the auto-pilot systems on the self-driving car as an example, the malicious user could conduct an adversarial attack on a road sign by querying his own vehicle model and then create an adversarial sticker to fool other vehicles using the same detection system.

Only given adversarial examples as evidence, in order to make the tracing possible, adversarial examples generated by different models have to be **unique** so that we are able to find the source model and trace the malicious user in the end. To achieve this goal, we design a random mask watermarking strategy which embeds the watermark to the generated adversarial samples without sacrificing model performance. At the same time, the proposed strategy is efficient and scalable that only needs a few iterations of fine-tuning. In the presence of the original example, a high-accuracy tracing method is proposed, which compares the adversarial perturbation with every model's masked pattern and the adversarial example's output distribution among different models. Because it is not always practical to have the original example as a reference, in the second part, we further discuss the most challenging and practical attack setting where only the adversarial example is available for the investigator. Observing that the model's probability predictions on the same adversarial example would change significantly with a different watermark applied, we derive an effective rule to find the compromised model. Specifically, based on the property that adversarial example is not robust against noise, we redesign the tracing metric based on the change in the predicted probabilities when applying different watermarks, which we expect the compromised model to minimize.

Comprehensive experiments are conducted on multiple adversarial attacks and datasets. When there is only a **single** adversarial example available, the results demonstrate that the two proposed methods could successfully trace the suspect model with over 74% accuracy on average with the data-limited case and data-free case. The tracing accuracy increases significantly to around 97% when there are two adversarial examples available.

Our contributions are summarized below:

- To the best of our knowledge, we are the first to propose a novel and scalable framework to make it possible to trace the compromised model by only using a single sample and its corresponding adversarial example.

- In the absence of samples used to generate adversarial examples, we further utilize the prediction difference of each model with different watermarks to identify the adversary without any requirement on the original sample.

- Extensive experiments were conducted to demonstrate the effectiveness of the proposed framework to trace the compromised model that the malicious users utilize to conduct different black-box adversarial attacks under different network architectures and datasets. We showed that the adversary could be traced with high accuracy in different scenarios, and the proposed framework has good scalability and efficiency.

## 2 RELATED WORK

**Adversarial Attack**    Since the discovery of adversarial example (Szegedy et al., 2014), many attack methods have been proposed. Roughly speaking, based on the different levels of information accessibility, adversarial attacks can be divided into white-box and black-box settings. In the white-box setting, the adversary has complete knowledge of the targeted model, including the model architecture and parameters. Thus, back-propagation could be conducted to solve the adversarial object by gradient computation (Goodfellow et al., 2015; Kurakin et al., 2017; Madry et al., 2018; Carlini & Wagner, 2017). On the other hand, the black-box setting has drawn much attention recently, where the attacker could only query the model but has no direct access to any internal information. Based on whether the model feedback would give the probability output, the attacks could be soft-

label attacks or hard-label attacks. In the soft-label setting, ZOO attack (Chen et al., 2017) first proposed using a finite difference to estimate the gradient coordinate-wise and then conducted the gradient descent. It was then improved by selecting a better prior distribution (Ilyas et al., 2018b) and compressing the search space (Tu et al., 2019). To further increase the query efficiency, instead of calculating the full gradient, gradient-sign-based methods (Liu et al., 2018) have been proposed, and the attacks could still achieve a good success rate. Also, random-search-based methods such as Square attack (Andriushchenko et al., 2020) and SimBA (Guo et al., 2019) have found that the attacks could be more successful in other domains such as frequency, and the square pattern could further improve the query efficiency. On the other hand, boundary attack (Brendel et al., 2017) first proposed the hard-label setting and used the random search method to find the adversarial attacks. It was then improved by Chen et al. (2020) by finding a better sampling prior. OPT attack (Cheng et al., 2018) and Sign-OPT attack (Cheng et al., 2020b), on the other hand, formalized the hard-label attack into an optimization framework and used the zeroth-order method to solve it.

**Watermarking**  Model watermarking is introduced in intellectual property protection on machine learning systems. It could be roughly divided into two categories: white-box and black-box watermarking, depending on the accessibility to the model and its parameters in order to extract the watermark. For the white-box watermarking, the first scheme for DNN (Uchida et al., 2017) tended to embed the watermark into the training process. Later, because of the extra capacity available in the state-of-art neural networks, DeepMarks (Chen et al., 2018) encoded the watermark to be a binary vector in the probability density function of trainable weights. However, it is not always convenient for the model owner to do the formal verification in the white-box watermarking, and the white-box watermark is also vulnerable to statistical attacks (Wang & Kerschbaum, 2019). Black-box methods are then proposed to solve the limitations mentioned above. DeepSigns (Darvish Rouhani et al., 2019) has done some improvement over Uchida et al. (2017) and proposed a framework to incorporate the black-box case. Utilizing backdoor attacks to do watermarking is another big trend and has drawn much attention recently (Zhang et al., 2018; Adi et al., 2018). Adversarial examples are used to enable extraction of the watermark without requiring model parameters (Chen et al., 2019; Le Merrer et al., 2020). However, the forensic investigation of adversarial examples has a major difference from the model watermarking as the model watermarking must be shown in the generated adversarial examples. Also, it requires the tracking to be attack-agnostic. In other words, the tracing mark should be robust across different attacks simultaneously. However, the current watermarking methods in (Zhang et al., 2018; Adi et al., 2018) are either attack-dependent or have limited effect on the generated adversarial examples, which is not suitable for this task.

**Forensic investigation in Machine Learning**  Although machine learning methods have already been used in forensic science (Carriquiry et al., 2019), there are a few studies on building trustworthy machine learning from a forensic perspective. Most papers focus on how to identify the model stealing attack by introducing the watermarking approaches to protect the intellectual property of the deep neural networks. That is to say, a unified and invisible watermark is hidden into models that can be extracted later as special task-agnostic evidence. However, to the best of our knowledge, we are the first paper to study the adversarial attack from a forensic investigation perspective.

## 3 PROBLEM SETTING

We formalize the identification of the compromised model in the owner-customer distribution setting (Zhang et al., 2021). The machine learning service provider (owner) is assumed to own $m$ copies of model $f_1, f_2, \ldots, f_m$ for the same $K$-way classification task trained using the same training dataset. As inference efficiency is critical in time-sensitive applications such as auto-pilot systems, these model copies are first encrypted for intellectual protection and security concerns and then distributed to the $m$ customers (users). Therefore, the customers only have black-box access to their own distributed model. In other words, the user $i$ could only query his own model $f_i$ to get the prediction results without any access to the internal information about the model.

Unfortunately, a malicious user (adversary) exists who aims to fool the whole system, including other users' models, by conducting black-box adversarial attacks. Specifically, let the malicious user's model copy to be $f_{att}$ (the *compromised model*). As he does not have access to query other users' models, he then chooses to perform black-box attacks to his copy $f_{att}$ to generate an adversar-

ial example $\boldsymbol{x}_{adv}$. As all model copies are trained with the same dataset for the same classification task, the generated adversarial example could successfully lead to the misclassification of other users' models. Our task is to find the compromised model $f_{att}$ from the model pool.

## 4 METHODOLOGY

If the same model copy is distributed to every customer, the generated adversarial examples from different users' models will be identical to each other, and it thus becomes impossible to trace the adversary by only giving adversarial examples. In the following section, we propose our framework which consists of two parts shown in Figure 2. First, we design a simple random mask watermarking method that would have a limited effect on the models' accuracy while embedding distinctive features in adversarial examples, distinguishing them from those generated from other models. We further design a tail and head mechanism to make the training process scalable and efficient. Depending on the availability of the original example as a reference, we then propose two detection scenarios to identify the adversary from adversarial examples.

### 4.1 RANDOM MASK WATERMARKING

Since we need to identify the compromised model from a large pool of customer copies, it requires us to assign a unique identification mark for every customer copy, and the mark should be reflected in the generated adversarial example.

In this section, we design a simple but effective method by applying a random watermark on each of the $m$ model copies. As shown in Figure 2, for each model copy $f_i (1 \le i \le m)$, we randomly select a set of pixels $\boldsymbol{w}^i$ as the *watermark* on the training samples. Formally, denote the input as $\boldsymbol{x} \in \mathbb{R}^{W \times H \times C}$. For every model $f_i$, we randomly generate a binary matrix $\boldsymbol{w}^i \in \{0, 1\}^{W \times H \times C}$ by sampling uniformly. We call the $\boldsymbol{w}^i$ *mask* for model $f_i$, deciding the set of masked pixels. When $\boldsymbol{w}^i_{a,b,c} = 1$ for a specific pixel $(a, b)$ at channel $c$, value is set to be 0; otherwise, when $\boldsymbol{w}^i_{a,b,c} = 0$, the original pixel value is not modified. That is to say, for every input $\boldsymbol{x}$, the input after the mask $\tilde{\boldsymbol{x}}$ on model $f_i$ would be $\tilde{\boldsymbol{x}}^i_{a,b,c} := \boldsymbol{x}_{a,b,c} \cdot (1 - \boldsymbol{w}^i_{a,b,c})$ for each pixel $(a, b)$ at each channel $c$. For simplicity, we use $\tilde{\boldsymbol{x}}^i = \boldsymbol{x} \odot (1 - \boldsymbol{w}^i)$ to denote the masked sample $\boldsymbol{x}_i$ in the whole paper, where $\odot$ represents the element-wise product.

Each input is first applied with the mask and then fed into the model in both the training and inference phases. To speed up the training process and make the pipeline scalable to thousands of users, we add each model with a few network layers as head part $h_i$. The output of the head part will directly feed into a shared tail model $t$. In other words, we have each model copy as

$$f_i(\boldsymbol{x}) = t(h_i(\boldsymbol{x})), \tag{1}$$

Specifically, in the pretraining phase, we first train a model without the watermark from scratch as the base model. Then each model copy is assigned with a unique model head for the added specific watermark and shares a big common tail inherited from the base model. During the fine-tuning process, we freeze the parameters in the tail and embed the watermark to the model by only fine-tuning the weights in the head part with a few epochs. Our following experiments will show it is sufficient to embed watermark to a few layers in DNNs without sacrificing model accuracy. To be noted, the proposed method could be easily adapted to the new user case as we could just create a unique watermark and fine-tune the new head with a few epochs.

Since the distributed model copy's detail is encrypted and not accessible in the black-box setting, users are unaware that their models and the generated adversarial examples are watermarked. Moreover, as the watermarks are generated uniformly at random, each user's model copy can be assigned a unique watermark even when the number of users is large. Furthermore, since deep neural networks have been shown to be robust against random noise, the proposed watermarking would have a very limited effect on the model's performance, as shown in Table 1, when the masking ratio $\|\boldsymbol{w}^i\|_1$ is not too large.

In this paper, we only show a straightforward random mask watermarking scheme, which could already provide us with satisfactory tracing accuracy without hurting the model performance. We leave the design of the better watermarking methods under this framework to future works.

### 4.2 DATA-LIMITED ADVERSARY IDENTIFICATION

With the watermarking scheme described in Section 4.1, we can exploit the information embedded in the watermarked adversarial example (and the corresponding original example) to identify the compromised model.

We first introduce the *data-limited* case where the corresponding original example $x$, on which the given adversarial example $x_{adv}$ is based, is available. A natural example of this setting is that the malicious user generates an adversarial sticker based on a road sign using his own self-driving car. As the self-driving system is built by the service provider and shared, the adversarial stickers would fool other passing vehicles. Then, as the investigator, we aim to trace the adversary by analyzing the road sign and the adversarial sticker. We now discuss the design of the detector in the data-limited case, which is inspired by the mechanism of adversarial attacks. Specifically, since the adversarial attack is formalized as an optimization problem, the adversary takes the gradient of the designed loss function $\mathcal{L}$ with respect to the input $x$ to find the most effective perturbation.

**Adversarial Perturbation:** Formally, for the model $f_i$, the gradient of the designed loss function $\mathcal{L}$ with respect to the given sample $x$ is

$$\frac{\partial \mathcal{L}(f_i(\tilde{x}))}{\partial x_{a,b,c}} = 0 \quad \text{if } w_{a,b,c}^i = 1, \tag{2}$$

Since the black-box attacks are designed to approximate the gradients used in the white-box attacks, we could expect that the approximated gradients at the masked pixels would have a value close to 0 or be smaller in magnitude than the other pixels. Based on this observation, since we have access to the original example $x$, we could calculate the adversarial perturbation $\delta = x_{adv} - x$. If the adversarial example is generated by the compromised model copy $f_{att}$, values in $\delta$ should be much smaller in those coordinates where $w^{att} = 1$. Therefore, given $x_{adv}$ and $x$, we thus calculate a score for each model by summing up the absolute values of the adversarial perturbation overall masked pixels (of the corresponding model), i.e.,

$$\delta^i = \sum_{a,b,c} w_{a,b,c}^i \odot |x_{adv} - x|_{a,b,c} \tag{3}$$

**Adversarial Stability:** Moreover, we also observe that the cross-entropy loss between the prediction output of adversarial examples and the ground-truth label of clean examples differs among different models. Since adversarial examples should be identical to original examples visually, the ground-truth label could be easily inferred. Specifically, if the adversarial example $x_{adv}$ is generated from model $f_i$, the cross entropy loss $\mathcal{L}_{CE}(f_i(x_{adv}), y)$ is smaller than $\mathcal{L}_{CE}(f_j(x_{adv}), y)$ if $f_j(x_{adv}) \neq y$, $\forall j \neq i$, where $y$ is the ground truth label of $x$. Intuitively, model $f_i$ would have the smallest confidence on the ground-truth label since some of the adversarial perturbation may be blocked by other models' watermarks. We then combine the two metrics and calculate the final score for each model. Then, we take the model with the smallest score as the compromised model, i.e.,

$$att \leftarrow \operatorname*{argmin}_{1 \le i \le m} (\delta^i + \alpha \mathcal{L}_{CE}(f_i(x_{adv}), y)) \tag{4}$$

### 4.3 DATA-FREE ADVERSARY IDENTIFICATION

The previously introduced data-limited detector requires access to the original example as a reference, which is not realistic in many scenarios. Therefore, in the following section, we relax this constraint and discuss the tracing under the most challenging yet realistic setting where the only evidence available is the generated adversarial example. We propose a data-free detector based on the different model outputs when applying different masks to the adversarial example.

Formally, for the given adversarial example $x_{adv}$, we first apply every model's watermark $w^i$, $i \in [m]$ to create a set of masked adversarial examples $\{\tilde{x}_{adv}^i\}_{i=1}^m$ where $\tilde{x}_{adv}^i = x_{adv} \odot (1 - w^i)$. We then feed the masked adversarial examples set to each model $f_i$ to get its probability output. For every model $f_i$, we get a probability output matrix $P^i := [f_i(\tilde{x}_{adv}^1)^T, \ldots, f_i(\tilde{x}_{adv}^m)^T] \in [0, 1]^{m \times K}$, where each element in $P^i$ is $P_{a,b}^i = [f_i(\tilde{x}_{adv}^a)]_b$ and $K$ is the number of classes.

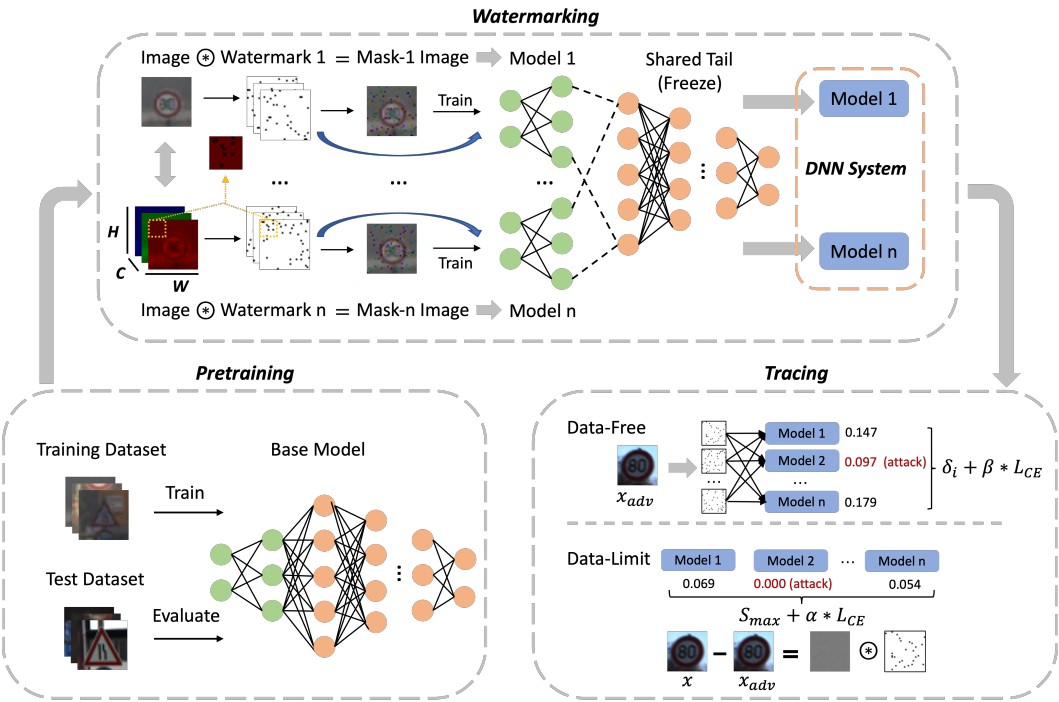

Figure 2: Proposed framework of identifying compromised model from adversarial examples.

Since adversarial examples are very close to the model's decision boundary (Brendel et al., 2017; Cheng et al., 2018), a slight perturbation to it would cause the model's prediction to change significantly. In other words, adversarial examples are sensitive to small perturbations, while ordinary examples are relatively more robust. It then inspires us to propose a metric based on this difference to detect the compromised model. Specifically, let us still assume the given adversarial example $x_{adv}$ is from model $f_i$. Then, when the corresponding watermark $w^i$ is applied, the probability prediction will remain unchanged. However, when applying another watermark $w^j, j \neq i$, it is likely that the watermarked adversarial example would be moved away from the decision boundary. Therefore, the maximal predicted class probability is generally larger after applying $w_j$. At the same time, if the adversarial example is not generated from the model, the extent of change would be limited. Therefore, we propose the max label score $S_{max}$ based on the extent of change of prediction:

$$S_{max}^i = \frac{\max_{1 \leq k \leq K} \boldsymbol{P}_{i,k}^i}{\sum_{1 \leq j \leq m} \max_{1 \leq k \leq K} \boldsymbol{P}_{j,k}^i} \tag{5}$$

We further combine the score of adversarial stability proposed in data-limited case with max label score to improve the detection accuracy:

$$att \leftarrow \underset{i}{\operatorname{argmin}}(S_{max}^i + \beta \mathcal{L}_{CE}(f_i(\boldsymbol{x}_{adv}), \boldsymbol{y})) \tag{6}$$

## 5 EXPERIMENTAL RESULTS

In this section, we extensively evaluate the proposed framework on a variety of adversarial attacks, datasets, and models. We first introduce the experiment setup and implementation details. In Section 5.1, we show the proposed random mask watermarking has a limited effect on the model performance. We then test the tracing success rate on adversarial examples generated by different black-box attacks in the data-limited case and data-free case in Section 5.2.

**Implementation Details:** We conduct our experiments on two popular image classification datasets GTSRB (Stallkamp et al., 2012) and CIFAR-10 (Krizhevsky et al., 2009). We use two widely used network architectures VGG16 (Simonyan & Zisserman, 2015) and ResNet18 (He et al.,

2016). All our experiments were implemented in Pytorch and conducted using an RTX 3090 GPU. In the pretraining stage, the base models are trained with Adam optimizer with the learning rate of $10^{-3}$ and a batch size of 128 for 50 epochs. For every model's watermark, we independently randomly sample 100 pixels to mask. As the input size of both CIFAR-10 and GTSRB is $32 \times 32 \times 3$, it is equivalent to saying the masking rate is around $3.26\%$. To embed different watermarks into individual models efficiently, we separate the base model by taking the first several layers as the head and leaving the rest as the tail. Specifically, for ResNet18, we take the first residual block as the model head, and we detach the first two convolution layers for VGG16. The parameter percentage of the head for ResNet18 and VGG16 is 1.34% and 0.26% respectively which is a very small portion of the whole network weights. We then fine-tune the weights in the head portion for 10 epochs while freezing the parameters of the model tail. Our code will be publicly available.

**Adversarial Attack Methods:** We perform the following five black-box adversarial attacks to generate the adversarial example:

- **Natural Evolutionary Strategy (NES):** Ilyas et al. (2018a) introduced a soft-label black-box adversarial attack that designed a loss on the output probability changes and used Neural evolution strategy (NES) to approximately estimate the gradient.

- **Bandits and Priors Attack (Bandit):** Ilyas et al. (2018b) introduced a soft-label black-box attack by using the bandit algorithm to find a better prior where the adversarial perturbation could be drawn with high probability.

- **Simple Black-box Attack (SimBA):** Guo et al. (2019) introduced a soft-label black-box adversarial attack by sampling the perturbation direction from a predefined orthonormal basis. The sampled direction would be either added or subtracted from the target image to test its success.

- **Hop-Skip-Jump Attack (HSJ):** Chen et al. (2020) introduced a hard-label black-box attack which is applied the zeroth-order sign oracle to improve Boundary attack (Brendel et al., 2017).

- **Sign-OPT Attack (SignOPT):** Cheng et al. (2020b) introduced hard-label black-box attack that use a single query oracle to improve the query efficiency of OPT attack (Cheng et al., 2018).

For SimBA, HSJ and SignOPT attacks, we use Adversarial Robustness Toolbox (ART) (Nicolae et al., 2018)'s implementation. We use the default hyperparameters in the ART toolbox to conduct the attack. For the NES attack and Bandit attack, we re-implement them in Pytorch following the official implementation in `https://github.com/MadryLab/blackbox-bandits`. All the attacks are conducted in the $\ell_2$ constraints and untargeted setting. The attack will be stopped when there is a successful adversarial example generated.

**Evaluation Metric:** To evaluate the effectiveness of the proposed detection method, for each attack, we generate 10 **transferable** adversarial examples for every model copy. An adversarial example $x_{adv}$ is defined as **transferable** if and only if the prediction of the compromised model $f_{att}$ is wrong and, at the same time, the prediction of at least one of the other $m - 1$ models is wrong. It is worth noting that it is rather difficult for the black-box attacks to have a good transferability over the proposed random mask watermarking so we consider 10 transferable examples are sufficient to test our detector's effectiveness. To sum up, for each attack, we have a total of 1000 adversarial examples under the setting of 100 models. We then define the tracing accuracy to evaluate the detection rate defined as follows:

$$\text{Trace Acc} = \frac{N_{\text{correct}}}{N_{\text{total}}} \tag{7}$$

Where $N_{\text{correct}}$ is the count of the correct identification of the compromised model and $N_{\text{total}}$ is the total number the transferable adversarial example generated.

## 5.1 MODEL PERFORMANCE WITH RANDOM MASK WATERMARKING

In this section, we conduct experiments to verify whether the model could still maintain a good performance after applying the watermark. Specifically, we train 100 models on two datasets CIFAR-10

and GTSRB with two popular architectures VGG16 and ResNet18. We also add a baseline model without watermark as a reference.

Table 1: The classification accuracies (%) of models with random mask watermarking. V-CIFAR10 represents the model trained with VGG16 using the CIFAR-10 dataset and R-GTSRB represents the ResNet18 model trained using the GTSRB dataset.

| Task | Baseline | Min | Mean | Median | Max |
|---|---|---|---|---|---|
| V-CIFAR10 | 90.70 | 89.30 | 89.71 | 89.72 | 90.20 |
| R-CIFAR10 | 91.97 | 91.10 | 91.49 | 91.51 | 91.83 |
| V-GTSRB | 97.60 | 96.10 | 96.99 | 97.02 | 97.48 |
| R-GTSRB | 98.50 | 96.81 | 97.45 | 97.47 | 98.15 |

In Table 1, it could be clearly observed that the accuracy of the watermarked models has a similar performance compared with the baseline model. The worst accuracy drops are only around 1%, while both mean and median keep a very similar performance with the baseline. Concerning there exists randomness in the training procedure, the proposed watermarking method has a limited effect on the model performance.

Table 2: The tracing accuracies (%) in data-limited and data-free scenarios with only a single adversarial example available.

| Case | Task | Bandit | HSJ | NES | SignOPT | SimBA | Mean |
|---|---|---|---|---|---|---|---|
| Data-Limit | V-CIFAR10 | 48.2 | 93.4 | 84.2 | 55.4 | 85.3 | 73.30 |
| | R-CIFAR10 | 54.2 | 95.5 | 87.4 | 65.8 | 83.0 | 77.18 |
| | V-GTSRB | 42.1 | 98.7 | 86.3 | 56.9 | 91.0 | 75.00 |
| | R-GTSRB | 43.8 | 98.7 | 86.3 | 61.8 | 86 | 75.32 |
| Data-Free | V-CIFAR10 | 66.2 | 83.9 | 71.6 | 85.7 | 59.2 | 73.32 |
| | R-CIFAR10 | 69.3 | 89.4 | 77.8 | 90.5 | 56.4 | 76.68 |
| | V-GTSRB | 62.4 | 92.0 | 67.5 | 90.7 | 56.3 | 73.78 |
| | R-GTSRB | 61.8 | 92.8 | 73.2 | 91.5 | 52.7 | 74.40 |

## 5.2 IDENTIFICATION RESULTS

For identification in the data-limited setting, we conduct experiments on 100 copies of models applied with random masks. We set the hyperparameter $\alpha$ to 0.85 for CIFAR10 and 0.5 for GTSRB and test tracing accuracy on different attacks. The results in the top half of Table 2 illustrate that our detection method could identify the compromised model successfully in all datasets and network architectures which achieves an average of 75.2% tracing accuracy with only one adversarial example. Specifically, our detection method is extremely effective to trace HSJ attack, where the average tracing accuracy is over 96.5%. However, it could also be seen that our tracing method has a poor performance on Bandit attack that achieves over 42% accuracy in the worst case, which may be affected by the noisy gradient estimation in the dimension reduction process. However, it could be solved by introducing a better watermark design, which we leave as our future work.

As we further limit the accessibility, we trace the compromised model with only one adversarial example and show the tracing accuracy at the bottom half of Table 2. For the data-free case, we also set the hyperparameter $\beta$ to 0.5 for both datasets. Although the original example is no longer available, we could still achieve a similar or even better tracing accuracy against some attacks. Specifically, the average tracing accuracy is 74.55% among different datasets and model architectures which is slightly lower than the data-limited case. However, we could see there is a significant tracing accuracy of around 20% improvement on the Bandit and SignOPT. It is because the data-free score is based on the property that the adversarial examples are more sensitive to a slight perturbation by different masks and more robust against noisy gradient estimation. Therefore, we could further combine those two scores in the practice to have a better tracing accuracy based on the available evidence.

**Results on adaptive attack**   To fully test the robustness of our proposed detectors, we also conducted an adaptive attack where the adversary has full access to the specific watermark embedded in each model. To be noted, it is not practical because users have only black-box access and it is not an easy task to directly infer which pixels are masked because of the noise estimation. The attacker then adds some Gaussian noise within the watermark to fool our tracing method. We test the average tracing accuracy across different noise levels on CIFAR10 with ResNet18 structure. Our results are shown in Table 3.

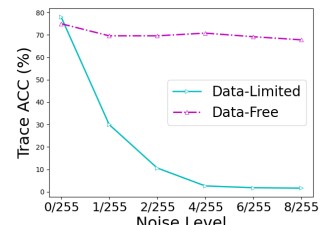

Figure 3: Average tracing accuracy on adaptive attack with different random noise levels.

Not surprisingly, we observe a significant accuracy drop in the data-limited case when adding random perturbation since we utilize the adversarial perturbation to identify the compromised model. However, we also notice that our data-free detector is not sensitive to random noise, which suggests that our tracing method can still be effective even if the adversary knows the predefined watermark.

### 5.3   RESULTS ON MULTIPLE ADVERSARIAL EXAMPLES

In the previous experiments, we considered only one adversarial example, which is the most extreme case for forensic investigation. However, here comes a natural question: could the proposed method have a better detection rate if more adversarial examples are collected? In this section, we conduct experiments to answer this question.

We use a simple strategy to combine multiple adversarial example scores. That is, we first calculate scores defined in Section 4.2 and Section 4.3 for each example, and then add up each score computed over all adversarial examples. Then we take the model with the smallest sum as the compromised model. We then conduct the experiments on 100 copies of the random mask watermarked ResNet18 and VGG16 models for the CIFAR-10 dataset in both the data-limited and data-free settings. It could be seen in Figure 4 that the detection rate keeps increasing with the number of adversarial examples. We could get around 97% tracing accuracy on average when adding only 1 adversarial example to current accessibility. And the accuracy will reach 100% if given three or more adversarial examples. It shows our method is quite robust and has a great potential to be further improved.

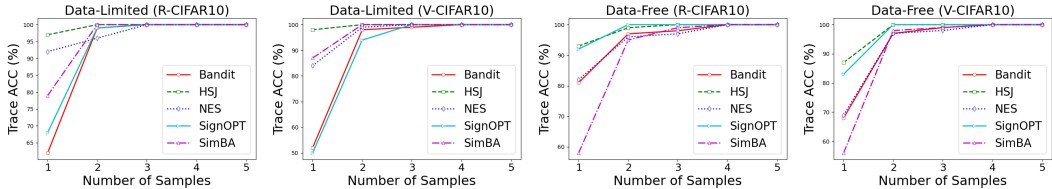

Figure 4: Tracing accuracy with different numbers of adversarial examples.

### 6   CONCLUSION AND LIMITATIONS

In this paper, we develop the first framework for identifying the compromised model from a single adversarial example for the forensic investigation. We first present a watermarking method to make the generated adversarial example unique and differentiable. Depending on the accessibility of the original example, two identification methods are presented and compared. Our results demonstrate that the proposed framework has a limited effect on the model's performance and has a high success rate to find the compromised model by only giving a single adversarial example. Our framework could further improve the detection rate to near 100% when two more adversarial examples are provided. As this study applied a simple yet effective random mask method to watermark models, future works could comprehensively study different watermarking methods in search of more efficient ways to distinguish the models in the setting of this study. Moreover, more experiments could be conducted to design even more effective quantitative scores to identify the adversary in future works.

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
