# OpenReview forum: "Identification of the Adversary from a Single Adversarial Example"
_ICLR.cc/2023/Conference — Submitted to ICLR 2023_

### Official Review · Reviewer_2gko · 2022-10-19

**Confidence:** 4
**Correctness:** 2
**Technical Novelty And Significance:** 2
**Empirical Novelty And Significance:** 2
**Recommendation:** 3

**Clarity, Quality, Novelty And Reproducibility:**

No code is attached to the paper for reproducibility.
The idea is nice and might be also novel, but it is not addressed properly.

**Strength And Weaknesses:**

**Strengths**
* the paper presents an interesting forensics problem, that can be useful to understand if malicious actors are trying to attack a specific model.

**Weaknessess**
* **Uniqueness is stressed but not proved.** While all the methodology is built on the intuition that watermarking creates unique copies that help analysers to trace either the culprit or the asset under attack, the authors do not analyse if the resulting adversarial examples are really unique among all of the models they use. Such limits the contribution of the paper, and the best suggestion would be a better rephrasing that help the reader in understanding that uniqueness may not be guaranteed in this schema.

* **Data-limited analysis is unclear.** The authors state that there might be cases where both the clean sample and its adversarial version are available, by bringing an example on autonomous driving. While the example per se is not clear to understand, the assumption of having both sample is strong, and not supported in the paper. Also, this would similar to detect the adversarial example and the original point, which is not an easy task (and the authors did not comment on that). The authors need to discuss more why this case is realistic.

* **Adaptive attack is not adaptive.** The authors should test the attack when the defense is known, hence the masking is more accurate than just adding random gaussian noise. The provided experiment is needed, since it is good to show the effect of attack at different capabilities of the attacker, but also further experiments are needed.

* **Wrong or unsupported technical statements.** The paper presents statements that undermine the contribution since they are either false or only partially true.
    * It is not true that adversarial examples are close to the boundary. If an attack is a minimum norm / minimum distance [a,b,c] then it is true. Otherwise, if the attack is computed with a maximum confidence attack (i.e. maximising the misclassification error), the sample can be found deep inside the target distribution [d, e]. Authors should check [f] for transferability of adversarial examples.
    * Equation 2 is confusing and not supported by any proof.
    * the paper says that "since adversarial examples should be identical to original examples visually, ground-truth label could be easily inferred" is not supported. As specified before, this problem is neither trivial or easy to formulate. If that would be the case, the authors should discuss in detail the statement.
    * the authors do not provide any support for the CE loss to be higher / lower when scoring an adversarial example computed on one model or another (Adversarial stability paragraph)


[a] Carlini, N., & Wagner, D. (2017, November). Adversarial examples are not easily detected: Bypassing ten detection methods. In Proceedings of the 10th ACM workshop on artificial intelligence and security (pp. 3-14).

[b] Pintor, M., Roli, F., Brendel, W., & Biggio, B. (2021). Fast minimum-norm adversarial attacks through adaptive norm constraints. Advances in Neural Information Processing Systems, 34, 20052-20062.

[c] Rony, J., Hafemann, L. G., Oliveira, L. S., Ayed, I. B., Sabourin, R., & Granger, E. (2019). Decoupling direction and norm for efficient gradient-based l2 adversarial attacks and defenses. In Proceedings of the IEEE/CVF Conference on Computer Vision and Pattern Recognition (pp. 4322-4330).

[d] Madry, A., Makelov, A., Schmidt, L., Tsipras, D., & Vladu, A. (2018, February). Towards Deep Learning Models Resistant to Adversarial Attacks. In International Conference on Learning Representations.

[e] Biggio, B., Corona, I., Maiorca, D., Nelson, B., Šrndić, N., Laskov, P., ... & Roli, F. (2013, September). Evasion attacks against machine learning at test time. In Joint European conference on machine learning and knowledge discovery in databases (pp. 387-402). Springer, Berlin, Heidelberg.

[f] Demontis, A., Melis, M., Pintor, M., Jagielski, M., Biggio, B., Oprea, A., ... & Roli, F. (2019). Why do adversarial attacks transfer? explaining transferability of evasion and poisoning attacks. In 28th USENIX security symposium (USENIX security 19) (pp. 321-338).


**Summary Of The Paper:**

The authors propose an identification mechanism that understands which model has been tested with adversarial examples, by leveraging a random-based watermarking technique that is kept inside the malicious points.
The methodology is tested in a black-box scenario, where the attacker can only retrive the score of a target classifier, but they can not access the weights or the data of the model.
Experiments are conducted in two ways: data-limited analysis refers to the usage of both the original sample and the adversarial one, while data-free analysis refers to tracing the compromised model with just one observation.
The authors show that the detection mechanism they implement, based on the scores of the pool of deployed models, is able to spot which one has been attacked in both cases.
The authors also test their attack in the presence of an adversary that tries to mimic the watermark by applying random gaussian noise.

**Summary Of The Review:**

Technical contribution is not proved, part of the threat model can be better discussed to convey why it is realistic, experimental part regarding the adaptive attack is lacking, many technical flaws and unsupported statements.

---

> ### Author Response · Authors · 2022-11-13
> **Responses Part 1/2**
>
> Thanks for your constructive feedback, we carefully address your concerns below.
>
> **Concern 1**: Uniqueness is stressed but not proved. While all the methodology is built on the intuition that watermarking creates unique copies that help analysers to trace either the culprit or the asset under attack, the authors do not analyse if the resulting adversarial examples are really unique among all of the models they use. Such limits the contribution of the paper, and the best suggestion would be a better rephrasing that help the reader in understanding that uniqueness may not be guaranteed in this schema.
>
> Thanks for your comment. The uniqueness of adversarial examples generated by different models is the result of the specific watermark embedded in the individual model. Since the masked pixels have zero contribution to the loss, the adversarial attack would naturally ignore those regions so that the generated adversarial example with different model copies will be unique. As our first metric used in the data-limited case, we utilize the pixel difference between the original image and the adversarial image within the mask to distinguish the compromised model from a bunch of candidate models, where the uniqueness of mask location ensures such identification is feasible. We have made it more clear in the revision.
>
> **Concern 2**: Data-limited analysis is unclear. The authors state that there might be cases where both the clean sample and its adversarial version are available, by bringing an example on autonomous driving. While the example per se is not clear to understand, the assumption of having both sample is strong, and not supported in the paper. Also, this would similar to detect the adversarial example and the original point, which is not an easy task (and the authors did not comment on that). The authors need to discuss more why this case is realistic.
>
> Thanks for your comment. In this paper, we focus the aftermath of an adversarial attack so that we have already known the attack has been conducted. Similar with crime forensic investigation, we could achieve the adversarial examples from different traces by investigating some potential remains from the environment or figuring out the time when the attack is conducted. It is thus not difficult for us to find the original input if the adversarial has been found by checking other similar inputs. Therefore, we don’t need to detect adversarial example from original examples. For example, in the autonomous driving, the malicious generate an adversarial sticker based on the stop sign and paste in the stop sign. After the accident happens, it is natural for the investigator to investigate and know the accident is caused by the misclassification on the stop sign. Therefore, the stop sign with the adversarial stick would be recorded as the adversarial example and it would be easy to get the original example by using the standard stop sign picture or just peeling off the sticker.
>
> However, we also realize sometimes it may not be easy for the investigator to get both examples. That’s why in the next section, we also design the identification method for the data-free case where only adversarial examples are available with satisfactory tracing accuracy. Therefore, even if the data-limited case is not supported, we could still utilize the method proposed in the data-free case to solve such tracing problem.
>
> *TO BE CONTINUED*

---

> > ### Author Response · Authors · 2022-11-13
> > **Responses Part 2/2**
> >
> > Thanks for your constructive feedback, we carefully address your remaining concerns below.
> >
> > **Concern 3**: Adaptive attack is not adaptive. The authors should test the attack when the defense is known, hence the masking is more accurate than just adding random Gaussian noise. The provided experiment is needed, since it is good to show the effect of attack at different capabilities of the attacker, but also further experiments are needed.
> >
> > Thanks for your comment. In the adaptive attack, we assume the attacker could somehow have some knowledge on the mask. Therefore, we add Gaussian noise to the pixels **within** the watermark instead of injecting a global gaussian noise. By adding noise to the specific region, the data-limited method's performance would be affected because we utilize the pixel difference between the original and adversarial example as a metric for identification. However, our data-free method shows less performance degradation which could alleviate the threat brought by the adaptive attack.
> >
> > **Concern 4**: Wrong or unsupported technical statements. The paper presents statements that undermine the contribution since they are either false or only partially true.
> >
> > 1. We adopt an early-stop strategy when generating adversarial examples, we stop the optimization process once the adversarial attack is successful which makes the adversarial examples close to the boundary.
> > 2. Equation 2 in our paper illustrates an ideal case where the gradients are explicitly computed with full access to model parameters. The gradient with respect to the masked pixel will be zero because those masked pixels have zero contribution to the classification loss both in the training and inference time. In other words, the loss won’t change when the masked pixel changes so there is no back-forward gradient flow on the specific pixels within the watermark.
> > 3. The major requirement for an adversarial example is humans would have correctly classified the adversarial example however the model will be fooled into a wrong classification, which means the contextual meaning of the examples shouldn’t be changed by the adversarial attack. Therefore, we believe, as an investigator, it is natural that the ground truth label could be inferred directly.
> > 4. To address your concern, we simply utilize the CE loss as the metric and perform the tracing task on the Tiny-Imagenet dataset. The tracing accuracies (%) are shown below:
> >
> > ||Bandit| HSJ |NES |SIgnOPT |SimBA|
> > |  ----  | ----  | ----  | ----  |  ----  | ----  |
> > |ResNet18 (data-limited) |60 |74 |58 |69 |62|
> > |VGG16 (data-limited) |51 |55 |46 |52 |49|
> >
> > The results demonstrate that with only CE loss as the metric, our tracing method could still achieve a satisfactory result.
> >
> > *END OF FIRST RESPONSE*

---

### Official Review · Reviewer_fDWu · 2022-10-21

**Confidence:** 5
**Correctness:** 2
**Technical Novelty And Significance:** 2
**Empirical Novelty And Significance:** Not applicable
**Recommendation:** 3

**Clarity, Quality, Novelty And Reproducibility:**

The paper is easy to read.
The novelty of the proposed watermark technique is limited. The approach as described very close to the works related to pixel dropouts.

[1] Cognitive Data Augmentation for Adversarial Defense via Pixel Masking, Pattern Recognition Letters (PRL), 2021
[2] Dropping pixels for adversarial robustness, IEEE CVPRW, (2019)

There is no mention of releasing the code to the community.

**Strength And Weaknesses:**

+ The idea of identifying the network vulnerable to adversarial attack is interesting and can be explored to develop defenses.

- The vulnerability of the technique against adaptive attacks is a major concern. In literature, several attacks have been developed to showcase the ineffectiveness of existing defense works; which only makes defense a challenging task.
- The limitation can also be seen from the progress of several effective transferable attacks and hence an attacker might not be utilizing one of the models the owner is sharing.



**Summary Of The Paper:**

The paper proposes an adversarial identification technique in a multi-model owner-customer environment setup. The authors have incorporated the watermark-based technique to uniquely identify the network used in the generation of adversarial examples. The watermark somewhat needs to be present in the adversarial example for its unique identification.

**Summary Of The Review:**

While the idea of the paper is interesting; several limitations need attention:

1. The novelty as described in the section above is limited and needs justification along with comparison and contrast with the existing works. How many masks per network are generated and what is the impact of these masks? Are these watermark masks random, fixed, or optimized through the network?

2. Are these masks unique for each attack or need to be different for different attacks?

3. The assumption that an attacker will utilize one of the networks might not be ideal in the real world. Due to the tremendous success of black-box and transferable, it is hard to detect from which networks the images are generated if a different surrogate model or no model information has been used.

Crafting Adversarial Perturbations via Transformed Image Component Swapping," in IEEE Transactions on Image Processing, 2022

4. The success of tracing in Table 2 is close to random (50%) for most of the attacks, making the success of this approach questionable.

5. The adaptive attacks can seriously break the proposed defense.

6. Limited evaluation of networks, datasets, and no SOTA attacks makes the proposed study a shallow work.

---

> ### Author Response · Authors · 2022-11-13
> **Responses**
>
> Thanks for your constructive feedback, we carefully address your concerns below.
>
> **Concern 1**: The novelty as described in the section above is limited and needs justification along with comparison and contrast with the existing works. How many masks per network are generated and what is the impact of these masks? Are these watermark masks random, fixed, or optimized through the network?
>
> Thanks for your comment. For the owner-customer distribution setting, we randomly generate one unique mask corresponding to each individual network. The clean accuracy comparison between watermarked model and the base model is shown in Table 1 which demonstrates there is no explicit impact brought by the watermarking strategy.
>
> **Concern 2**: Are these masks unique for each attack or need to be different for different attacks?
>
> Thanks for your comment. The mask is unique for individual models despite different attacks. The mask uniqueness for different networks is the cornerstone of our method since we highly rely on the unique information embedded in the adversarial example to identify the real compromised model. Therefore, there is no direct correlation between mask uniqueness and different attacks.
>
> **Concern 3**: The assumption that an attacker will utilize one of the networks might not be ideal in the real world. Due to the tremendous success of black-box and transferable, it is hard to detect from which networks the images are generated if a different surrogate model or no model information has been used.
>
> Thanks for your comment. The transferable setting (e.g., use a surrogate model) is out of our consideration in this paper and we would leave the problem to future work.
>
> **Concern 4**: The success of tracing in Table 2 is close to random (50%) for most of the attacks, making the success of this approach questionable.
>
> Thanks for your comment. In our setting, we aim to identify the compromised model out of 100 candidate models where the probability for the random guess should be 1% instead of 50%.
>
> **Concern 5**: The adaptive attacks can seriously break the proposed defense.
>
> Thanks for your comment. The adaptive attack could break our proposed data-limited case since we utilize the pixel difference between the original and adversarial example as a metric for identification. However, it is under extreme conditions that we assume the attacker could somehow have some knowledge of the mask. Meanwhile, our proposed data-free method shows robustness to the adaptive attack which could alleviate the threat brought by the adaptive attack. In the practice, both data-limited and data-free methods could be jointly used to further improve the detection performance.
>
> **Concern 6**: Limited evaluation of networks, datasets, and no SOTA attacks makes the proposed study a shallow work.
>
> Thanks for your comment. As far as we have considered, our paper designs the first framework to identify the compromised model in a forensic view with only one adversarial example. And we believe our work would serve as a baseline to further develop effective tracing methods.
>  All black-box attacks are widely used to evaluate the model’s adversarial robustness. HSJ and SignOPT are considered as the SOTA hard-label blackbox attack.
>
> To address your concern, we further evaluate our method on the Tiny-ImageNet dataset, and the experimental results are shown in the following table:
>
> | |Bandit | HSJ | NES | SignOPT | SimBA |
> |  ----  | ----  | ----  | ----  |  ----  | ----  |
> | ResNet18 (data-limited) | 54.8 | 81.0| 65.6| 53.4| 85.1 |
> | VGG16 (data-limited) | 47.5 | 64.1 | 51.4 | 54.2 | 78.7 |
> | ResNet18 (data-free) | 58.5 |72.3 |59.2| 70.1| 62.4 |
> | VGG16 (data-free) |55.2 | 62.1 |48.4 |58.7 |56.5 |
>
> Compared with previous tracing results on CIFAR-10 and GTSRB, our experiments on Tiny-ImageNet for both cases (data-limited and data-free) demonstrate our tracing strategy is still effective with only a slight drop in the overall detection accuracy. We have also included these results in the revision.

---

### Official Review · Reviewer_wEyj · 2022-10-23

**Confidence:** 4
**Correctness:** 3
**Technical Novelty And Significance:** 2
**Empirical Novelty And Significance:** 2
**Recommendation:** 5

**Clarity, Quality, Novelty And Reproducibility:**

The idea proposed in the paper is quite original and novel and can help for forensic investigation when machine learning systems under attack. However, watermarking all the models with the proposed approach requires a significant effort form a computational perspective that may be limiting for deployments with many users/devices and large machine learning models.
The intuition about the proposed method is clear. However, as I’ll explain below there are certain aspects that can be improved, both in the proposed method and the experimental evaluation.


**Strength And Weaknesses:**

Strengths:
+ The idea of trying to identify the model that has been compromised by an adversary using the proposed scheme with watermarking for forensic investigation is quite original and novel.
+ The method is fairly intuitive and the paper is written and organized.

Weaknesses:
+ The paper lacks a more thorough analysis of the capabilities and limitations of the proposed scheme (see below) and to test the robustness against more advanced adaptive attacks (the one considered is a bit naïve).
+ In the empirical results show that, in some cases, the data-free approach (with less knowledge for the defender) is less effective than the data-limited approach. This shows limitations on the proposed algorithms for the data-limited approach and suggest that the authors can easily improve the detection rate by considering also the score used for the data-free detection in equation (5).
+ The watermarking technique is perhaps too simple, which I think that is limiting the performance of the proposed detection method.


**Summary Of The Paper:**

The paper introduces a novel technique for detecting which model has been compromised (i.e. an adversarial example has been created targeting that specific model) from a set of deployed models that have been watermarked. Thus, each model deployed has a watermark so that some of the inputs are masked. When an attacker tries to compromise one of these models, the adversarial example contains some of the distinct features of the watermark introduced, which helps to detect the origin of the adversarial example. The authors consider two models: one where the defender knows the original and the adversarial example (data-limited adversary identification) and another one where the defender only knows the adversarial example (data-free adversary identification). The authors tested their proposed method on VGG16 and ResNet-18 models on CIFAR-10 and GTSRB datasets.

**Summary Of The Review:**

Overall, I think that the idea explored by the authors is very interesting. However, the paper still requires more work:
+ The watermarking method proposed, as well as the detection techniques methods are simple and very intuitive. However, it looks like the adversary can also take advantage of this for evading detection. For instance, if the attacker is aware of the watermark scheme (not necessarily having access to the masks used to watermark the models), the attackers can also introduce bogus information in pixels/features where they think that are part of the mask used for the target model by looking at the perturbation introduced for each pixel. Alternatively, attackers can also play with the confidence level of the crafted adversarial examples, which may difficult the detection by the proposed algorithm. However, the adaptive attack used in the evaluation is too simple (it just adds Gaussian noise) and does not necessarily exploit the weaknesses of the proposed method. Thus, a more thorough analysis on the robustness against adaptive attackers is necessary.
+ Related to the previous point, for instance, it is not clear to me that the proposed method can be robust against black box attacks generated with procedural noise functions or structure noise patterns, as the one proposed by Co et al. “ Procedural Noise Adversarial Examples for Black-Box Attacks on Deep Convolutional Networks” (CCS, 2019), where the masking of some pixels will be possible ignored by the attack.
+ The watermarking scheme is possibly too simple, which limits the performance of the proposed method. It is also not very clear, how this can perform in typical computer vision datasets (e.g. ImageNet) with a lot more pixels. On the other hand, the detection performance for some of the attacks is not very good (for a single example). I believe this can be improved with a better watermarking scheme.
+ As mentioned before, it is odd that the data-limited adversary identification algorithm performs worse than the data-free one in some cases. This suggests that the first algorithm can be improved (by for example using the score in equation (5)). On the other side, the combination of the different elements in the detection algorithms is not explored properly in the experiments. Thus, the parameters alpha and beta are cherry picked (or no explanation is given about how to select them). In this sense, a sensitivity analysis (ablation studies) would be necessary, as well as a mechanism to select these hyperparameters.

---

> ### Author Response · Authors · 2022-11-13
> **Responses Part 1/2**
>
> Thanks for your constructive feedback, we thoroughly considered your comments and addressed your concerns below.
>
> **Concern 1** However, watermarking all the models with the proposed approach requires a significant effort form a computational perspective that may be limiting for deployments with many users/devices and large machine learning models.
>
> Thanks for your comment. As claimed in the paper, to solve the potential computational efficiency problem with a large number of models, we adopt the fine-tuning strategy where we separate the whole model into head and tail parts, and only fine-tune the parameters in the head with only several epochs while freezing the parameter in the tail. The training time is only 108.90 seconds for training a VGG16 head and 155.25 seconds for a ResNet18 head on an RTX 3090. At the same time, the fine-tuning process could be easily parallelized, and we believe the whole procedure is efficient.
>
> **Concern 2**: The watermarking method proposed, as well as the detection techniques methods are simple and very intuitive. However, it looks like the adversary can also take advantage of this for evading detection. For instance, if the attacker is aware of the watermark scheme (not necessarily having access to the masks used to watermark the models), the attackers can also introduce bogus information in pixels/features where they think that are part of the mask used for the target model by looking at the perturbation introduced for each pixel. Alternatively, attackers can also play with the confidence level of the crafted adversarial examples, which may difficult the detection by the proposed algorithm. However, the adaptive attack used in the evaluation is too simple (it just adds Gaussian noise) and does not necessarily exploit the weaknesses of the proposed method. Thus, a more thorough analysis on the robustness against adaptive attackers is necessary.
>
> Sorry for the possible confusion about the adaptive attack. In the adaptive attack, we assume the attacker could somehow have some knowledge of the mask. Therefore, we add Gaussian noise to the pixels within the watermark instead of injecting a global gaussian noise. By adding noise to the specific region, the data-limited method's performance would be affected because we utilize the pixel difference between the original and adversarial example as a metric for identification. However, our data-free method shows less performance degradation which could alleviate the threat brought by the adaptive attack.
>
> **Concern 3**: Related to the previous point, for instance, it is not clear to me that the proposed method can be robust against black box attacks generated with procedural noise functions or structure noise patterns, as the one proposed by Co et al. “ Procedural Noise Adversarial Examples for Black-Box Attacks on Deep Convolutional Networks” (CCS, 2019), where the masking of some pixels will be possible ignored by the attack.
>
> Thanks for your comment. The experiments are under preparation.
>
> *TO BE CONTINUED*

---

> > ### Author Response · Authors · 2022-11-13
> > **Responses Part 2/2**
> >
> > Thanks for your constructive feedback, we addressed your remaining concerns below.
> >
> > **Concern 4**: The watermarking scheme is possibly too simple, which limits the performance of the proposed method. It is also not very clear, how this can perform in typical computer vision datasets (e.g. ImageNet) with a lot more pixels. On the other hand, the detection performance for some of the attacks is not very good (for a single example). I believe this can be improved with a better watermarking scheme.
> >
> > Thanks for your comment. As the first framework for compromised model identification, our method utilized a simple watermark technique to embed unique information within an individual model. While our watermark strategy is simple, the experimental results show the effectiveness of our method. We also believe the masking method could be further improved (e.g., using an optimized mask generator) on the identification performance.
> >
> >  To evaluate the tracing effectiveness in typical computer vision datasets, we extend our experiments on the Tiny-ImageNet dataset with the results shown in the following table:
> >
> > | |Bandit | HSJ | NES | SignOPT | SimBA |
> > |  ----  | ----  | ----  | ----  |  ----  | ----  |
> > | ResNet18 (data-limited) | 54.8 | 81.0| 65.6| 53.4| 85.1 |
> > | VGG16 (data-limited) | 47.5 | 64.1 | 51.4 | 54.2 | 78.7 |
> > | ResNet18 (data-free) | 58.5 |72.3 |59.2| 70.1| 62.4 |
> > | VGG16 (data-free) |55.2 | 62.1 |48.4 |58.7 |56.5 |
> >
> > While the tracing accuracy drop is observed, our method could still achieve 63.58% and 60.34% in the data-limited case and data-free case respectively, demonstrating our tracing method's effectiveness even with a more complex dataset (scenario). We have included these results in the revision.
> >
> > **Concern 5**: As mentioned before, it is odd that the data-limited adversary identification algorithm performs worse than the data-free one in some cases. This suggests that the first algorithm can be improved (by for example using the score in equation (5)). On the other side, the combination of the different elements in the detection algorithms is not explored properly in the experiments. Thus, the parameters alpha and beta are cherry picked (or no explanation is given about how to select them). In this sense, a sensitivity analysis (ablation studies) would be necessary, as well as a mechanism to select these hyperparameters.
> >
> > Thanks for your comment. The case where the data-limited adversary identification algorithm performs worse than the data-free one is caused by hyper-parameter alpha in the data-limited detection loss function. We made a sensitivity analysis on Tiny-ImageNet with the VGG16 in the data-limited case. The tracing accuracies (%) with 100 models are shown in the table below:
> >
> > |alpha | Bandit| HSJ |NES |SignOPT| SimBA
> > |  ----  | ----  | ----  | ----  |  ----  | ----  |
> > |0.5 |16 |82 |61 |20 |87
> > |0.8 |20 |80 |61 |28 |85
> > |1 |28 |80 |58 |30 |85
> > |2 |38 |73 |55 |37 |83
> > |5 |46 |64 |52 |53 |79

---

> > > ### Comment · Reviewer_wEyj · 2022-11-30
> > > **Comments after rebuttal**
> > >
> > > Thank you very much for your responses. After reading all the comments from the other reviewers and the authors' replies, I still think that there are aspects in the paper that require more work. Thus, I have decided to keep my previous score.

---

### Official Review · Reviewer_RQg3 · 2022-10-25

**Confidence:** 3
**Clarity, Quality, Novelty And Reproducibility:** The paper is well written and the met…
**Correctness:** 3
**Technical Novelty And Significance:** 4
**Empirical Novelty And Significance:** 3
**Recommendation:** 6

**Strength And Weaknesses:**

The paper is well written.

The application scenarios described in the paper are interesting and novel to me.

The experiment results show that their method is effective.

Some concerns are as follows:
1. What about the performance on large-scale datasets? For instance, ImageNet.
2. Can the proposed method be applied to other tasks? For instance, detection and segmentation?

**Summary Of The Paper:**

The paper proposes a novel framework to trace the compromised model by only using a single adversarial example. The method can be applied in forensic investigation scenarios.

**Summary Of The Review:**

The work is novel and interesting, but perhaps needs more experiments.

---

> ### Author Response · Authors · 2022-11-13
> **Responses**
>
> Thanks for your positive feedback. We carefully address your concerns below.
>
> **Concern 1**: What about the performance on large-scale datasets? For instance, ImageNet.
>
> Thanks for your comments, we further evaluate our method on the Tiny-ImageNet dataset, and the tracing accuracies (%) are shown in the following table:
>
> | |Bandit | HSJ | NES | SignOPT | SimBA |
> |  ----  | ----  | ----  | ----  |  ----  | ----  |
> | ResNet18 (data-limited) | 54.8 | 81.0| 65.6| 53.4| 85.1 |
> | VGG16 (data-limited) | 47.5 | 64.1 | 51.4 | 54.2 | 78.7 |
> | ResNet18 (data-free) | 58.5	|72.3	|59.2|	70.1|	62.4 |
> | VGG16 (data-free)	|55.2 | 62.1	|48.4	|58.7	|56.5 |
>
> Compared with previous tracing results on CIFAR-10 and GTSRB, our experiments on Tiny-ImageNet for both cases (data-limited and data-free) demonstrate our tracing strategy is still effective with only a slight drop in the overall detection accuracy. We have included these results in the revision.
>
> **Concern 2**: Can the proposed method be applied to other tasks? For instance, detection and segmentation?
>
> Thanks for your valuable advice, as the first framework for identifying the compromised models, we only take the image recognition task to evaluate the effectiveness of our method. We would further explore other application scenarios and consider extending our proposed method to other computer vision tasks in our future work.

---

### Decision · Program_Chairs · 2023-01-20

**Decision:**

Reject

**Justification For Why Not Higher Score:**

The reviewers raise several problems and concerns, including evaluation of adaptive attack, analysis, performance, ans so on.
During the rebuttal process, the reviewers still did not reach a consistent consensus of giving positive comments on this paper.
Besides, the most common comment is that this paper needs more work to make it complete.

**Justification For Why Not Lower Score:**

N/A

**Metareview: Summary, Strengths And Weaknesses:**

In this paper, instead of enhancing adversarial robustness, the authors take the investigator's perspective and propose a new framework to trace the first compromised model by only using a single adversarial example in a forensic investigation manner.
Overall, the reviewers recognize that the intuition is clear, the idea of identifying the network vulnerable to adversarial attack is interesting, the paper is well-organized and is to understand, and contains some novelty.
However, the reviewers raise several problems and concerns, including evaluation of adaptive attack, analysis, performance, ans so on.
During the rebuttal process, the reviewers still did not reach a consistent consensus of giving positive comments on this paper.
Besides, the most common comment is that this paper needs more work to make it complete.
Therefore, this paper is recommended to be rejected.